# Comprehensive Profiling of Paper Mulberry (*Broussonetia papyrifera*) Crotonylome Reveals the Significance of Lysine Crotonylation in Young Leaves

**DOI:** 10.3390/ijms23031173

**Published:** 2022-01-21

**Authors:** Yibo Dong, Chao Chen

**Affiliations:** 1Department of Plant Protection, Institute of Crop Protection, College of Agriculture, Guizhou University, Guiyang 550025, China; 13511972016@163.com; 2College of Animal Science, Guizhou University, Guiyang 550025, China

**Keywords:** posttranslational modification, paper mulberry, plant metabolism, lysine crotonylation, PPI

## Abstract

Lysine crotonylation is a newly discovered and reversible posttranslational modification involved in various biological processes, especially metabolism regulation. A total of 5159 lysine crotonylation sites in 2272 protein groups were identified. Twenty-seven motifs were found to be the preferred amino acid sequences for crotonylation sites. Functional annotation analyses revealed that most crotonylated proteins play important roles in metabolic processes and photosynthesis. Bioinformatics analysis suggested that lysine crotonylation preferentially targets a variety of important biological processes, including ribosome, glyoxylate and dicarboxylate metabolism, carbon fixation in photosynthetic organisms, proteasome and the TCA cycle, indicating lysine crotonylation is involved in the common mechanism of metabolic regulation. A protein interaction network analysis revealed that diverse interactions are modulated by protein crotonylation. These results suggest that lysine crotonylation is involved in a variety of biological processes. HSP70 is a crucial protein involved in protecting plant cells and tissues from thermal or abiotic stress responses, and HSP70 protein was found to be crotonylated in paper mulberry. This systematic analysis provides the first comprehensive analysis of lysine crotonylation in paper mulberry and provides important resources for further study on the regulatory mechanism and function of the lysine crotonylated proteome.

## 1. Introduction

Histone posttranslational modifications (PTMs) are playing an important role in expanding the genetic code and regulating cellular physiology [1]. Post-translational modification of proteins refers to the post-translational chemical modification of proteins, which can regulate the active position and folding of proteins, as well as the reaction between proteins and other biological macromolecules (including nucleic acids and lipids of proteins), and plays an important regulatory role in many important life activities and diseases. More than 400 post-translational modifications have been identified, including ubiquitination, phosphorylation, acetylation, glycosylation, methylation and prenylation [2,3,4]. Among PTMs, a newly discovered lysine acylation named lysine crotonylation is associated with various disease states in mammals [5] and plays a vital role in nearly every biological process, including development and growth.

Histone crotonylation modification refers to the introduction of a crotonyl group onto the amino acid residue of histones. This is a novel histone acylation modification. Histone lysine crotonylation was first discovered in 2011, mainly related to active chromatin [6]. Crotonylation also occurs in the ε-amino group of the lysine residue and modifies the histone charge, but its four-carbon length and planar orientation differ from that of acetylation [7]. It has been shown that crotonyl modification can promote gene expression and maintain the activity of genes associated with meiosis metaphase. After the introduction of the negatively charged crotonyl group, the positive charge of histones is reduced, and the binding to the negatively charged DNA becomes looser, which is conducive to the binding of transcription factors. In addition, similar to acetylation, the substrate for crotonylation is a donor molecule linked to the sulfhydryl group of coenzyme A (CoA) via a thioester, named crotonyl CoA [8].

In proteome-wide lysine crotonylation studies, Liu et al. performed liquid chromatography–mass spectrometry/mass spectrometry analysis of mouse liver crotonylation under normal physiological conditions and identified 10,034 Class I (localization probabilities > 0.75) crotonylation sites among 2245 proteins using a label-free strategy [9]. Zhang et al. comprehensively evaluated the crotonylation proteome of Botrytis cinerea and identified 3967 Kcr sites among 1041 proteins. Bioinformatics analysis demonstrated most crotonylated proteins were distributed in the cytoplasm (35%), mitochondria (26%) and nucleus (22%) [10]. Kwon et al. performed the first proteome-wide analysis of Kcr in zebrafish larvae and identified 557 Kcr sites on 218 proteins, representing the Kcr event in zebrafish embryos. The results provide a reference for future studies on the effects of lysine crotonylation on aging and heart failure [11]. Yang et al. performed a lysine crotonylation quantification on Rhodotorula strains (*Rhodotorula mucilaginos*). Subsequently, 1691 lysine crotonylation sites were identified in 629 protein groups, among which we quantified 1457 sites in 562 proteins. Among the quantified proteins, 79 and 46 crotonylated proteins were upregulated and downregulated, respectively, and the differentially upregulated modified proteins were mainly involved in the tricarboxylic acid cycle and gluconeogenic pathway [12]. Sun et al. conducted a global crotonylation proteomic analysis of tobacco (*Nicotiana tabacum*) and identified a total of 2044 lysine modification sites distributed over 637 proteins, and lysine crotonylation was associated with various metabolic pathways including carbon metabolism, the citrate cycle, glycolysis and the biosynthesis of amino acids [13]. Studies have identified 2799 Kcr sites which were identified on 908 proteins with 14 conserved motifs in the Chinese mitten crab (*Eriocheir sinensis)*. Bioinformatics analysis showed that Kcr was mainly found on proteins in cytoplasm, mitochondria and nucleus, as well as on proteins in the endoplasmic reticulum involved in ribosomes, proteasomes, carbon metabolism and protein processing [14]. A study by Liu et al. identified 1265 lysine crotonylation sites on 690 proteins in rice (*Oryza sativa* L. *japonica*) seedlings, and the results provide a comprehensive understanding of the biological functions of the crotonylome and new active histone modifications in transcriptional regulation in plants [15]. However, lysine crotonylation has rarely been reported in woody plants.

Paper mulberry (*Broussonetia papyrifera*) is a deciduous tree or shrub that grows naturally in Asia and Pacific countries such as China, Thailand and the USA. Roots, bark and fruits are all used in traditional Chinese medicines [16], due to their wide distribution, fast growth, strong adaptability and resistance to extreme environments. Unlike mulberry (*Morus alba*, M.a.), paper mulberry can acclimate to Karst soil [17]. In addition, paper mulberry has the characteristics of light loving, strong adaptability, drought and barren resistance. Moreover, paper mulberry leaf feed has high nutritional value and is an important cash crop and an important tree species for urban and rural greening [18]. A previous study obtained a chromosome-scale genome assembly for paper mulberry using integrated approaches, including the Illumina and PacBio sequencing platforms, as well as Hi-C, optical and genetic maps. Comparative genomic analysis revealed expansion and contraction in the flavonoid and lignin biosynthetic gene families, explaining the reasons for increased flavonoid biosynthesis and decreased lignin biosynthesis in paper mulberry, respectively [19]. Previous studies have clarified that lysine crotonylation plays an important role in plant physiology [20]. The proteome-wide identification of lysine-crotonylated proteins has not been accomplished in paper mulberry, so the elucidation of lysine crotonylation in paper mulberry cells is important for understanding the role of lysine crotonylation in regulating development and stress responses.

In this study, we reported a qualitative study of crotonylated proteomics in paper mulberry leaves. A large-scale analysis of lysine crotonyl in paper mulberry leaves was carried out through crotonylation enrichment technology and the qualitative proteomics research strategy of high-resolution liquid chromatography-mass spectrometry. These findings have broadened our perception of lysine crotonylation involvement in protein regulation, and they will be instrumental for illustrations of the potential functions and regulatory metabolism of this newly identified PTM in woody plants.

## 2. Results and Discussion

### 2.1. Profiling the Crotonylsites and Crotonylproteins on Paper Mulberry Young Leaf Tissues

Lysine crotonylation (Kcr) is an evolution-conserved histone posttranslational modification (PTM) [21]. However, its regulatory role is poorly studied in the woody plant paper mulberry. To achieve an overview of lysine crotonylation in paper mulberry, we conducted systematic and qualitative crotonylated profiling in young leaf tissues of paper mulberry through four steps: protein extraction, trypsin affinity enrichment, LC–MS and bioinformatics analyses (Figure 1A). A total of 6600 peptides and 5158 crotonylated peptides were identified by spectrogram analysis. A total of 5195 crotonylation sites were identified on 2272 proteins (Figure 1B and Appendix A). A previous report identified 971 crotonylated proteins from the woody plant *Camellia sinensis* [22], however, in the current study, the number of lysine crotonylation sites was twice as many and the number of crotonylated proteins was twice as many as previously reported. The mass errors for those crotonyl peptides were near 0, with the majority less than 5 ppm, implying a high level of accuracy of the data. The length of most of the identified crotonylpeptides ranged from 7–22 amino acids with a few longer exceptions, even more than 30 amino acids (AAs) (Figure 1C). Over 50% of the crotonylpeptides carried only 1 crotonylsite; approximately 23% of the crotonylpeptides covered 2 crotonylsites; approximately 10% crotonylpeptides covered 3 crotonylsites, and approximately 10% of the remaining crotonylpeptides had more than three crotonylsites (Figure 1D). The abundance of lysine crotonylation in other plant samples was reviewed (Table 1), which may explain the significant differences in protein crotonylation levels between species. In addition, we speculate that the current study identified more acetylated peptides than previous studies, due in large part to improved antibody efficiency and mass spectrometry sensitivity.

### 2.2. Analysis of Crotonylated Lysine Motifs

To estimate the nature of lysine crotonylation sites in paper mulberry, we investigated the sequence motifs in the identified peptides with the Motif-x program (Massachusetts, CA, USA). We searched the potentially conserved motifs in the context of 21 AA-long sequences centered by crotonylsites. Twenty-seven conserved sequences, with amino acid sequences (fold increase > 1.5, *p* < 0.01) from 10 to +10 surrounding the crotonylated lysine, were extracted from 5158 crotonylated peptides (Figure 2A), and it is worth noting that motif preference was identified in proteins associated with different biological processes, molecular functions, pathways, and domains. These motifs were EKcrE, Kcr******EKcr, YKcr**E, FKcrE, K******HKcr, EKcr, KcrE, KcrD, GKcr, AKcr, FKcr, YKcr, NKcr, SKcr, HKcr, PKcr, DKcr, TKcr, WKcr, H**Kcr, VKcr, LK, GKcr, IKcr, MKcr, KKcrF and CKcr (Ksu indicates the lysine crotonylation site, and * represents an unspecified amino acid residue), and they exhibited different abundances (Figure 2A). In accordance with these findings, the heat map results showed that the frequencies of lysine A, D, E, K at multiple positions, lysine F at +1 position, lysine G at +1, −1 positions and lysine H at +1 and −1 positions were overrepresented around the modified lysine sites, including lysine N at −1 position, lysine R at +6 position, lysine V at +1 position and lysine Y at −1, −2, −3 and +4 position (Figure 2B). Most of these amino acids are of a positive charge. The amino acid preference may reflect a bona fide preference or may be due to the preference of antibodies used for selective enrichment of the crotonylated peptides. Two crotonylation motifs, EKcr and DKcr, were also found in tea, rice and papaya [15,22,23], indicating that E and D may be important conservative residues in croton acylation in plant proteins, which also confirmed that lysine crotonylation is a highly conserved posttranslational modification in various species. Therefore, proteins with specific amino acid residues in the corresponding positions are more likely to be modified by crotonyl groups in paper mulberry.

### 2.3. Functional Characterization of Lysine Crotonylome

To better understand the function of the substrates of lysine crotonylation in paper mulberry, we classified all of the crotonylated proteins to a gene ontology (GO) functional classification analysis based on biological process, cellular component and molecular function (Figure 3 and Appendix A).

The GO functional classification of all the acetylproteins was investigated in terms of biological process, cellular component and molecular function. The results of biological process analysis showed that most of the crotonylated proteins were involved in the metabolic process (25%), response to stimulus (19%) and cellular process (13%) (Figure 3A), which accounted for over half of the identified crotonylated proteins, indicating a primary role of lysine crotonylation in metabolism. It is also worth noting that response to stimulus accounted for 19%, mainly including response to abiotic stimulus, response to endogenous stimulus and response to external stimulus. This may also be the reason why the paper mulberry can grow well under the environmental conditions of karst lime soil. For the “molecular function” category, most crotonylated proteins were found to be related to binding (54%) and catalytic activity (26%) in the molecular function classification (Figure 3B), suggesting that massive crotonylated proteins are involved in DNA transcription or protein interaction. Consistent with these findings, most crotonylated proteins were found to be associated with organelles, cells and membranes in the cellular component classification, accounting for 40%, 21% and 10% of all the identified proteins, respectively (Figure 3C). A large proportion of the identified crotonylated proteins in paper mulberry leaves were located in the chloroplast (43%), cytoplasm (24%) and nucleus (16%) (Figure 3D) and were shown in subcellular localization analysis, is consistent with previous research results [24], suggested that chloroplast metabolism-related enzymes may be the target and substrate of lysine crotonylation. Eukaryotes have long been thought to have arisen by evolving a nucleus, endomembrane and cytoskeleton [25].

In summary, lysine crotonylated proteins are widely distributed in different pathways, indicating that this posttranslational modification plays an important role in cell metabolism.

### 2.4. Functional Enrichment Analysis of Crotonylated Proteins

For the annotation of all identified proteins containing the modification site and the screening of the corresponding proteins of the modification site, to detect whether the modification showed a significant enrichment trend in some functional types, we conducted enrichment analysis at three levels: GO classification, KEGG pathway and protein domain (Figure 4). The results of biological process enrichment show that most crotonylated proteins play important roles in metabolic processes and photosynthesis (Figure 4A, red bar, Appendix A). Consistent with these observations, many modified proteins were found to be associated with enzyme activity and binding activity in molecular function enrichment analysis (Figure 4A, blue bars, Appendix A). In agreement with these findings, proteins located in the cytoplasm, chloroplast, plastid and thylakoid were more crotonylated based on cell component enrichment analysis (Figure 4A, green bars, Appendix A). The carboxylation of methylcrotonyl-CoA is catalyzed by a new biotin enzyme, methylcrotonyl-CoA carboxylase (MCC), which can be inhibited by avidin. It is not present in isolated chloroplasts but seems to be a component of plant mitochondria [26].

Similar results were obtained in the enrichment analysis of protein domains and KEGG pathways. The KEGG pathway results showed that most crotonylated proteins were related to ribosome, glyoxylate and dicarboxylate metabolism, carbon fixation, citrate cycle (TCA cycle) and glycolysis/gluconeogenesis (Figure 4B and Appendix A), suggesting that crotonylation is involved in basic metabolic regulation in paper mulberry.

Enriched protein domain analysis showed the proteasome subunit, PsbP, thioredoxin, glutathione S-transferase, N-terminal domain, glutathione S-transferase, C-terminal domain, glutathione S-transferase, C-terminal domain, starch synthase catalytic domain, oxidoreductase NAD-binding domain, ATP synthase alpha/beta family, nucleotide-binding domain, etc. were enriched for crotonylated protein domain enrichment (Figure 4C and Appendix A). The crotonallin/26S proteasome pathway has been implicated in diverse aspects of eukaryotic cell regulation through its ability to rapidly remove intracellular proteins. Thioredoxins are ubiquitous disulfide reductases that regulate the redox status of target proteins. Plant thioredoxins display striking diversity not found in other organisms [27], suggesting that proteins related to enzyme metabolism may be largely crotonylated in paper mulberry. ATP synthase is an essential component of the photosynthetic apparatus in chloroplasts and is one of the major protein complexes of the thylakoid membrane [28].

Therefore, lysine crotonylated proteins are widely distributed in different pathways, indicating that this posttranslational modification regulates a variety of cellular metabolic processes.

### 2.5. Crotonylated Proteins Involved in Photosynthesis and Carbon Fixation

In paper mulberry, photosynthesis is one of the most important metabolic processes. To deeply investigate the photosynthesis regulated by lysine crotonylation, we examined the accumulation levels of crotonylation proteins involved in photosynthesis and the carbon fixation pathway (Figure 5). In this study, the results of GO annotation (Figure 4A) and KEGG pathway analysis (Figure 4B) also suggest that lysine crotonylation plays an essential role in photosynthesis and the carbon fixation pathway. Several crotonylated proteins have been identified in photosystems I and II, the cytochrome b6/f complex and photosynthetic electron transport. Figure 5A shows that five protein subunits of photosystem I (PSI) (PsaD, PsaE, PsaF, PsaG, PsaN), four protein subunits of photosystem II (PSII) (PsbQ, PsbR, PsbS, Psb27), one subunit of the cytochrome b6/f complex (Pet C), two subunits of photosynthetic electron transport (PetE, PetH) and three subunits of F-type ATPase were among the vital crotonylated proteins. In addition, in photosystem II in photosynthesis, the ability of plant chloroplasts affected by external stimulate to use carbon dioxide to assimilate carbon is limited under external stress conditions, such as drought and high-temperature stress, thereby reducing energy consumption [29], the proportion of electrons transferred to O_2_ is relatively increased; thus, O_2_ and H_2_O_2_ can be formed [30]. Under the catalysis of metal ions, more reactive and aggressive -OH can be formed [31]. In the face of oxidative stress caused by drought, paper mulberry can provide rich antioxidants for stress proteins. Through the biological process of photosynthesis, plants fix solar energy and atmospheric carbon in the plants. The products of photosynthesis, such as ATP and NADPH, then participate in carbon fixation. As such, we further investigated the succinylated proteins involved in carbon fixation in paper mulberry. A total of 23 enzymes were identified to participate in the carbon fixation pathway in photosynthesis (Figure 5B). Among these enzymes, 10 were related to glycolysis/gluconeogenesis: triosephosphate isomerase, phosphoglycerate kinase 3, fructose-bisphosphate aldolase 2, phosphoenolpyruvate carboxykinase (ATP), glyceraldehyde-3-phosphate dehydrogenase, fructose-1,6-bisphosphatase, pyruvate kinase, cytosolic isozyme and NADP-dependent glyceraldehyde-3-phosphate dehydrogenase. Furthermore, four enzymes were involved in glyoxylate and dicarboxylate metabolism (Figure 5C). Malate dehydrogenase 1, ribulose bisphosphate carboxylase small chain and ribulose bisphosphate carboxylase large chain OS were also found to be crotonylated. These results indicate that paper mulberry Kcr proteins have close interactions in photosynthesis and glyoxylate and dicarboxylate metabolism; these enzymes involved in carbon metabolism regulate carbon fixation, energy transduction and photosynthetic efficiency through lysine crotonylation.

### 2.6. Interactive Networks among Crotonylated Proteins in Paper Mulberry

Function prediction based on the protein interaction network has become a hot spot in protein function analysis and prediction, which can integrate various data information and has the advantage of measuring and predicting protein functions accurately at an overall level [32]. To further investigate the role of crotonylation in cell regulation, we used algorithms from Cytoscape software [33] to establish a protein interaction network (Figure 6 and Appendix A). A total of 1569 crotonylation sites were identified as nodes in the Protein Interaction database, revealing a global view of the diverse cellular functions of crotonylated proteins in paper mulberry. Based on the Cytoscape algorithm, 30 clusters of crotonylated proteins were retrieved from the complicated interaction network. As shown in Figure 6, six highly interconnected clusters of crotonylated proteins were involved in citrate cycle, glutathione metabolism, glyoxylate and dicarboxylate metabolism, carbon fixation in photosynthetic organisms and ribosomes and proteasomes The results of the protein interaction network analysis demonstrated the involvement of crotonylation in the regulation of critical biological processes.

The interactions and network interactions between crotonylated proteins are extremely complex. Some crotonylated proteins are located at the nodes of the interaction network, reflecting that six biological processes are cross-linked, and crotonylated proteins are responsible for coordinating these cross-links. The complicated interaction networks of crotonylated proteins indicate that the physiological interactions between these protein complexes substantially improve their coordination and cooperation in this important woody plant.

### 2.7. Overlap between Lysine Crotonylation, Succinylation and Ubiquitylation in Paper Mulberry

In addition to crotonylation, other posttranslational modifications, including succinylation, ubiquitination, malonylation and acetylation, also occur on lysine residues [34]. To review whether there is a link between crotonylation and other lysine modifications, we compared the crotonylation proteins identified in this study with proteins from succinylated and ubiquitinated proteins of paper mulberry. We compared these modified proteins with the 2272 crotonylated proteins identified in this study and found that 408 paper mulberry proteins were modified by crotonylation, succinylation and ubiquitination (Figure 7A). By comparing the sites of these modified proteins, it was found that the three modified proteins had a total of 355 sites, of which 524 sites were both crotonylated and succinylated, and 671 sites were both ubiquitinated and crotonylated (Figure 7B and Appendix A). It is worth noting that the growth environment of the paper mulberry in this study is a karst landform environment, and drought and high temperature are important environmental stresses in the growth process of paper mulberry. We specifically analyzed the crotonylated proteins that may directly respond to stress. Based on the annotation, HSP70 (heat shock protein 70) was identified (Figure 7C). A representative heat shock protein 70 (HSP70) is shown in Figure 6. The Hsp70 molecular chaperones of plants are encoded by a multigene family, whose members are regulated in the process of plant growth and development, and differentially expressed under temperature stress and other conditions that interrupt normal protein folding or are beneficial to protein degeneration [35], playing vital roles in heat stress adaptation and protein degradation [36]. HSP70 exists in three isoforms in plants: a cytoplasmic form [37], a chloroplastic form [38] and a nuclear form [39]. Transient silencing of the hsp70 gene in tomato (*Solanum Lycopersicum* L.) was performed using reverse genetics to assess different growth and physiological parameters under normal conditions and during the response to drought stress. The results showed that hsp70 silencing led to severe growth retardation, death, significant membrane damage and leakage, decline in relative water content and a low rate of pigment accumulation [40].

The same finding was confirmed in Arabidopsis and Hevea brasiliensis [41]. These results suggest that HSP70 is a molecular chaperone in response to various environmental stresses, that high levels of HSP70 confer drought stress tolerance and that HSP70 plays an essential role in helping cells cope with adverse environments. The diverse and critical roles of heat shock proteins in plant metabolism suggest the possibility that the three different modifications may have different roles, especially for lysine crotonylation. Previous findings found that lysine crotonylation could positively regulate plant low-temperature stress response and enhance cold resistance [42]. In contrast, our study suggests that lysine crotonylation may also be involved in regulating plant high temperature stress response, which may also provide further evidence for the gene expression mechanism during plant high-temperature stress response.

## 3. Materials and Methods

### 3.1. Collection and Preparation of Plant Materials

The seedlings of paper mulberry (*Broussonetia papyrifera*) Zhong Ke 1 were grown in a greenhouse with the temperature set at 26/18 °C (day/night) and a photoperiod of 16/8 h (light/dark). This may also be the reason why the paper mulberry can grow well under the environmental conditions of karst lime soil. Three biological replicates of 15 g of leaves were harvested from 7-week-old seedlings for protein extraction.

### 3.2. Protein Extraction

Paper mulberry leaves were ground in liquid nitrogen, then the powder was transferred to a 5 mL centrifuge tube and subjected to three times on ice using a high-intensity ultrasonic processor (Scientz, Ningbo, China) in lysis buffer (including 1% Triton X-100, 10 mM dithiothreitol, 1% protease inhibitor cocktail (Calbiochem, Darmstadt, Germany), 50 μM PR-619, 3 μM TSA, 50 mM NAM and 2 mM EDTA) sonication. An equal volume of Tris-saturated phenol (pH 8.0) was added; then, the mixture was further vortexed for 5 min. After centrifugation (4 °C, 10 min, 5000× *g*), the upper phenol phase was transferred to a new centrifuge tube. The proteins were precipitated by adding at least 4 times the volume of ammonium sulfate-saturated methanol and incubated at −20 °C for at least 6 h. After centrifugation at 4 °C for 10 min, the supernatant was removed. The remaining precipitate was washed once with ice-cold methanol and then three times with ice-cold acetone. The protein was redissolved in 8 M urea, and the protein concentration was determined with a BCA kit according to the manufacturer’s instructions.

### 3.3. Trypsin Digestion

For digestion, the protein solution was reduced with 5 mM dithiothreitol for 30 min at 56 °C and alkylated with 11 mM iodoacetamide for 15 min at room temperature in the dark. 100 mM TEAB was then added to urea at a concentration of less than 2 M to dilute the protein samples. Finally, trypsin was added at a trypsin-to-protein mass ratio of 1:50 for the first overnight digestion and a trypsin-to-protein mass ratio of 1:100 for the second 4 h digestion.

After trypsin digestion, peptides were desalted and vacuum dried using a Strata X C18 SPE column (Phenomenex, Tianjin, China). Peptides were reconstituted in 0.5 M TEAB solution and processed according to the manufacturer’s TMT kit/iTRAQ kit protocol. Briefly, one unit of TMT/iTRAQ reagent is thawed and reconstituted in acetonitrile. The peptide mixture was then incubated for 2 h at room temperature and mixed, desalted and dried by vacuum centrifugation.

### 3.4. Panantibody-Based PTM Enrichment

To enrich the crotonylated peptides, we dissolved the tryptic peptides in NETN buffer (pH 8.0) containing 1 mM EDTA, 100 mM NaCl, 0.5% NP-40 and 50 mM Tris-HCl and incubated them with prewashed antibody beads (PTM-402, PTM Bio, Hangzhou, China) at 4 °C with gentle shaking overnight. The beads were washed four times with NETN buffer and twice with H_2_O. The bound peptides were eluted with 0.1% trifluoroacetic acid, and the eluted fractions were combined and dried by centrifugal vacuum. Before LC-MS/MS analysis, the peptides were desalted using C18 zipport (Millipore, MA, USA) according to the manufacturer’s instructions.

### 3.5. LC-MS/MS Analysis

Tryptic peptides were dissolved in 0.1% formic acid (solvent A) and loaded directly onto a homemade reversed-phase analytical column (15-cm length, 75 μm inner diameter). The gradient consisted of solvent B (0.1% formic acid, 98% acetonitrile) increasing from 6% to 23% in 26 min, from 23% to 35% in 8 min, rising to 80% in 3 min, and then remaining at 80% for the final 3 min, all on an EASY-nLC 1000 UPLC system at a constant flow rate of 400 nL/min. The peptides were treated with an NSI source and then subjected to tandem mass spectrometry (MS/MS) in a Q-exclusive TM Plus (Thermo, MA, USA) chromatography coupled on-line to the UPLC [43]. The applied electrospray voltage was 2.0 kV. full scan *m*/*z* scans ranging from 350 to 1800 were used and intact peptides were detected in the Orbitrap at a resolution of 70,000. Peptides were then selected for MS/MS using an NCE setting of 28, and fragments were detected in Orbitrap at a separation of 17,500. This is a data-dependent procedure with 20 MS/MS scans followed by 15.0 s dynamic exclusion after one MS scan. Automatic gain control (AGC) was set to 5E4. The first fixed mass was set to 100 *m*/*z.*

### 3.6. Database Search

The MS/MS data obtained were processed using the MaxQuant (MPIB, Berlin, Germany) search engine (v.1.5.2.8). Tandem mass spectra were searched against the paper mulberry protein database concatenated with the reverse decoy database. Trypsin/P was designated as a cleavage enzyme allowing up to 4 missing cleavages. The mass tolerance for precursor ions was set as 20 ppm in the first search and 5 ppm in the main search, and the mass tolerance for fragment ions was set as 0.02 Da. Carbamidomethyl on Cys was designated as a fixed modification, and variable modification and oxidation on Met were designated as variable modifications. The FDR was adjusted to <1%, and the minimum score for modified peptides was set >40.

### 3.7. Bioinformatics Methods

The gene ontology (GO) annotation proteome was obtained from the UniProt-GOA database (http://www.ebi.ac.uk/GOA/) (accessed on 16 December 2021) [44]. Then, proteins were classified by gene ontology annotation based on three major categories: biological process, cellular component and molecular function. The protein pathways were annotated using the Kyoto Encyclopedia of Genes and Genomes (KEGG) database [45]. Then, the annotation result was mapped to the KEGG pathway database using the KEGG online service tool KEGG mapper. InterPro (http://www.ebi.ac.uk/interpro/) (accessed on 16 December 2021) is a database that integrates various information about protein families, structural domains and functional sites and is available through a web-based interface and service. Wolfpsort is an updated version of PSORT/PSORT II for the prediction of eukaryotic sequences. For the specificity of prokaryotic species, subcellular localization prediction soft CELLO was used [46]. All differentially expressed modified protein database accessions or sequences were searched against the STRING database version 10.5 for protein–protein interactions [47]. Only interactions between the proteins belonging to the searched dataset were selected, thereby excluding external candidates. STRING defines a metric called “confidence score” to define interaction confidence; we extracted all interactions with a confidence score >0.7 (high confidence). Sequence models consisting of amino acids at specific positions of modify-21-mers (10 amino acids upstream and downstream of the site but modify-13-mers phosphorylation of 6 amino acids upstream and downstream of the site) in all protein sequences were analyzed using Soft MoMo (motif-x algorithm) [48]. All the database protein sequences were used as background database parameters. The minimum number of occurrences was set to 20. Check the simulation of the original motif-x, other parameters set as the default.

## 4. Conclusions

Through this investigative study, 5195 lysine crotonylation sites were identified on 2272 proteins of developing paper mulberry seeds, significantly larger than those reported in plants for acetylome and crotonylome. Amino acid position preferences for crotonylation sites were obtained by motif and flaking sequence analyses. A total of 27 distinguished motifs were identified. A comprehensive analysis of lysine crotonylome and protein modifications in paper mulberry seedlings was performed. In addition, extensive characterization of the crotonylome in paper mulberry indicates that crotonylation mainly occurred in proteins involved in multiple functions, ranging from control of metabolic processes to biological regulation. We found remarkable overlap between the crotonylated sites and the succinylated and ubiquitylated sites reported. In particular, lysine crotonylation of proteins associated with photosynthesis was revealed, indicating other possible important roles in regulating physiological processes. The results revealed that a large number of Kcr proteins in paper mulberry leaves participated in metabolic pathways, including croton-mediated proteolysis, catechin biosynthesis, carbohydrate and amino acid metabolism and other metabolic pathways.

## Figures and Tables

**Figure 1 ijms-23-01173-f001:**
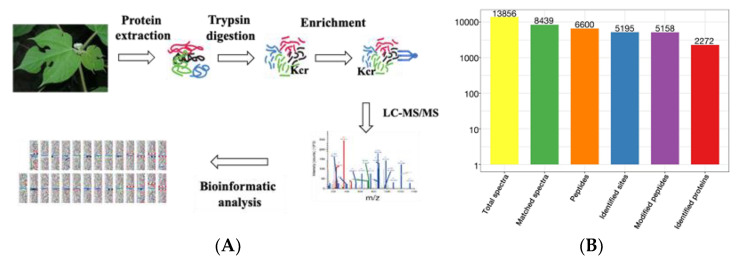
Identification of global crotonylation sites and proteins in paper mulberry leaves. (**A**) Experimental workflow for the identification of lysine crotonylome. (**B**) Basic statistical table of MS results. (**C**) Number of each identified peptide length. (**D**) Number of each identified modified site in a protein.

**Figure 2 ijms-23-01173-f002:**
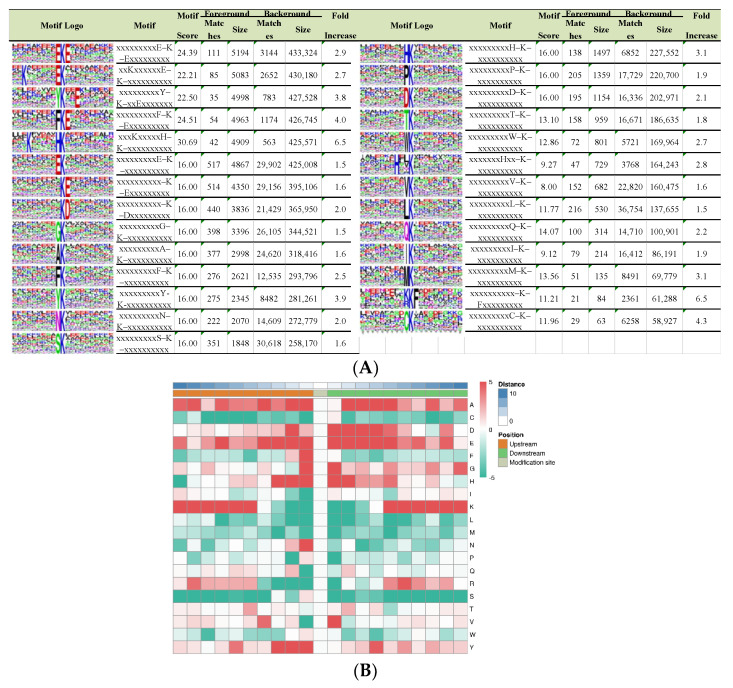
Properties of lysine-crotonylated peptides. (**A**) Crotonylation sequence motifs for ±10 amino acids around the lysine crotonylation sites. (**B**) Heat map of the amino acid compositions of the crotonylated sites.

**Figure 3 ijms-23-01173-f003:**
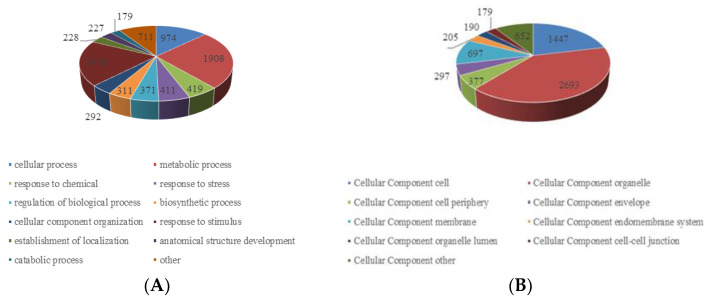
Gene ontology functional characterization of the identified crotonylated proteins. (**A**) Distribution of the crotonylated proteins in terms of biological processes. (**B**). Distribution of the crotonylated proteins in terms of cellular components. (**C**) Distribution of the crotonylated proteins in terms of molecular functions. (**D**) Subcellular location prediction.

**Figure 4 ijms-23-01173-f004:**
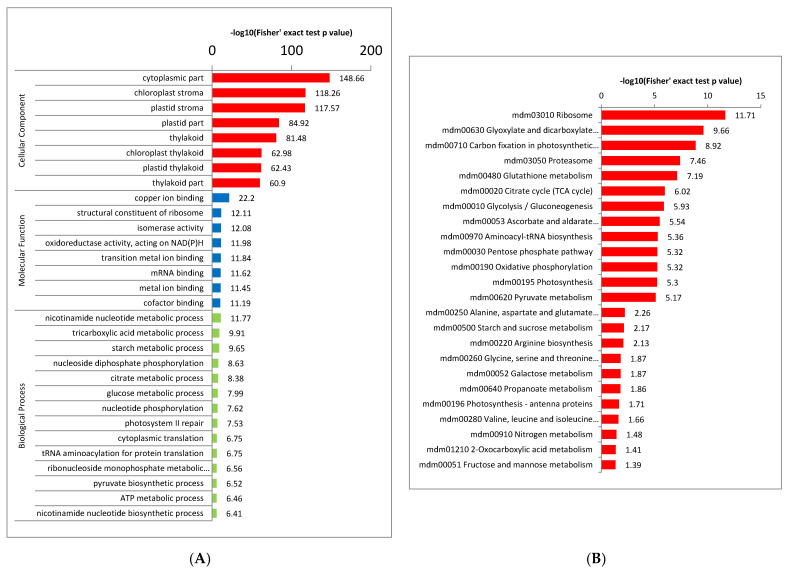
Enrichment analysis of the lysine-acetylated proteins in paper mulberry. (**A**) GO-based enrichment analysis in terms of cell component (red bars), molecular function (blue bars) and biological process (green bars). (**B**) KEGG pathway enrichment analysis. (**C**) Proteins identified as crotonylproteins in this study.

**Figure 5 ijms-23-01173-f005:**
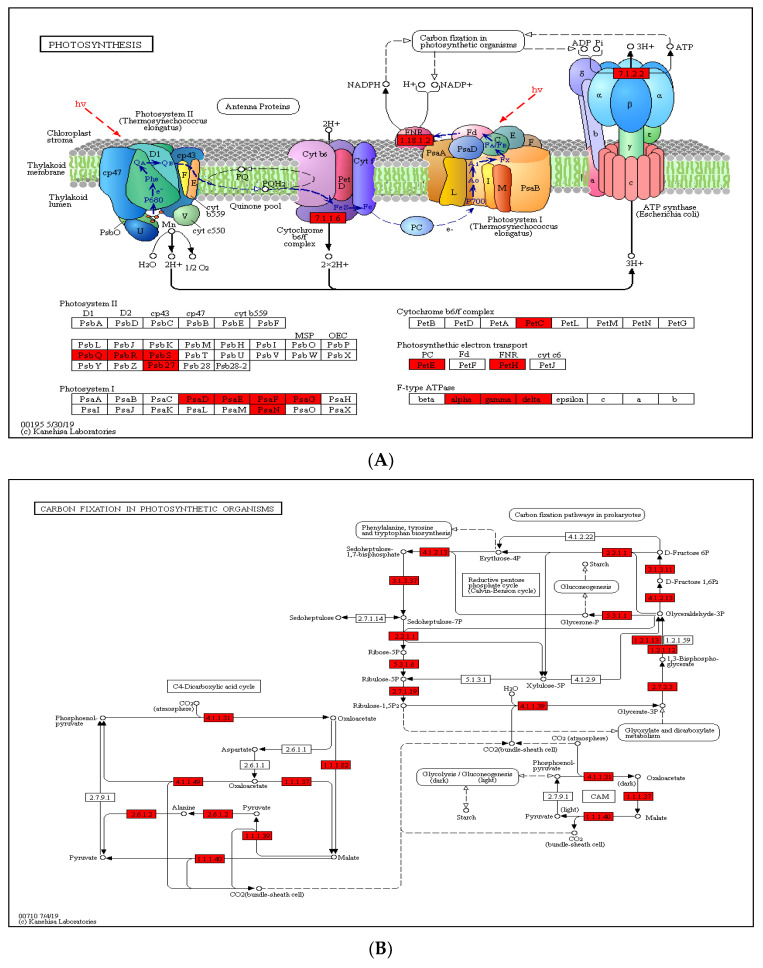
(**A**) Crotonylated proteins involved in photosynthesis. The identified crotonylated proteins are highlighted in red. (**B**) Crotonylated proteins involved in carbon fixation in photosynthetic organisms. The identified crotonylated proteins are highlighted in red. (**C**) Crotonylated proteins involved in glyoxylate and dicarboxylate metabolism. The identified crotonylated proteins are highlighted in red.

**Figure 6 ijms-23-01173-f006:**
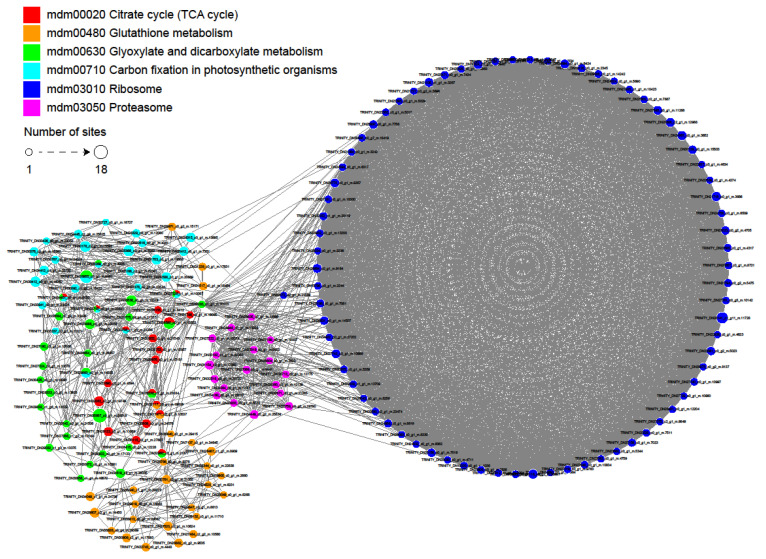
Interaction networks of the crotonylated proteins in paper mulberry.

**Figure 7 ijms-23-01173-f007:**
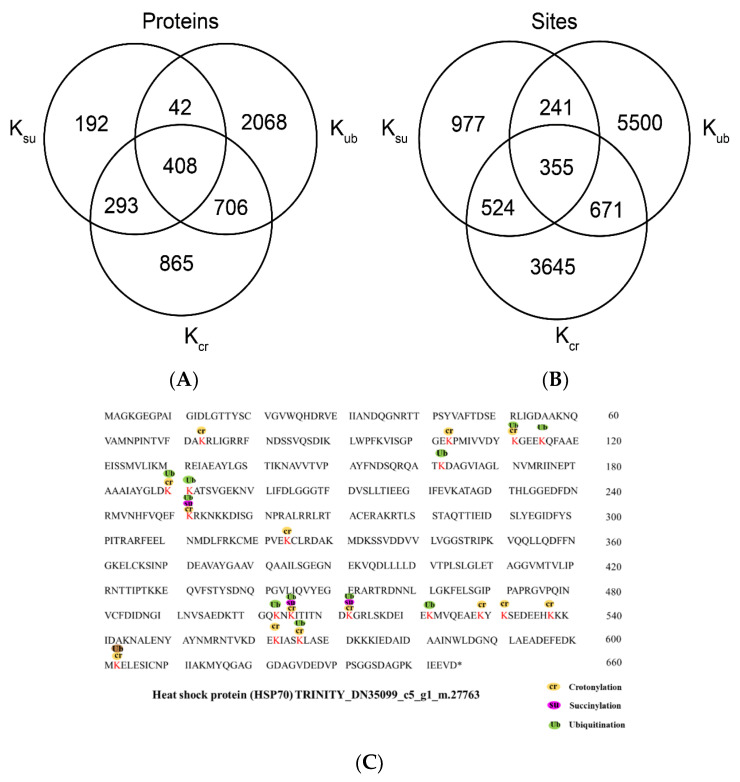
Analysis of proteins modified by crotonylation, succinylation, and ubiquitylation. (**A**) Overlap of crotonylated (Kcr) proteins in this study with reported paper mulberry succinylated (Ksu) and ubiquitylated (Kub) proteins. (**B**) Overlap of crotonylated lysine residues in this study with reported succinylated (Ksu) and ubiquitylated (Kub) lysine residues in paper mulberry. (**C**) Analysis of heat shock protein (HSP70) modified by crotonylation, succinylation and ubiquitination.

**Table 1 ijms-23-01173-t001:** Lysine crotonylome identified in paper mulberry and other plants.

Lysine Crotonylation	Number of Identified Proteins	Number of Identified Sites	Reference
Paper mulberry (*Broussonetia papyrifera*)	2272	5159	This study
Papaya (*Carica papaya* L.)	2120	5995	[23]
Peanut (*Arachis hypogaea* L.)	2508	6051	[13]
*Tobacco (Nicotiana tabacum)*	637	2044	[7]
Rice (*Oryza sativa* L. *japonica*)	690	1265	[15]

## Data Availability

All relevant data can be found within the manuscript and its supporting materials.

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
