# Peer review of "Comprehensive Profiling of Paper Mulberry (Broussonetia papyrifera) Crotonylome Reveals the Significance of Lysine Crotonylation in Young Leaves"

_ijms, 2022, doi:10.3390/ijms23031173_

Round 1
Reviewer 1 Report
Thr work entittled "Comprehensive Profiling of Paper Mulberry (Broussonetia pa-2 pyrifera) Crotonylome Reveals the Significance of Lysine Cro-3 tonylation in Young Leaves" by Dong and collaborators is an interesting work about one emerging PTM, lysine crotonylation, in plants. Despite description of results are fine and clear, figures are presented in a low-quality form and must be improved. Most of the times graph legends are lacking or illegible. Also, authors should perform improved versions of Figure 2 and Figure 6 to facilitate the understanding of results contents.
My main concerns are about novelty of reported results. Novelty of the work is only based on the fact of performing the analyses in another plant sp. (Broussonetia pa-2 pyrifera). However, others plant works also reached very similar conclusions. Authors do not mention nor discuss anything (in fact discussion section is missing) about this and in my opinion this is a major gap in the work. Authors must state clearly the main advances in comparison to previous works.
Reviewer 2 Report
Journal: IJMS (ISSN 1422-0067)
Manuscript ID: ijms-1534931
Type: Article
Title: Comprehensive Profiling of Paper Mulberry (Broussonetia papyrifera) Crotonylome Reveals the Significance of Lysine Crotonylation in Young Leaves
Authors: Yibo Dong , Xiaomao Wu , Chao Chen *
Section: Molecular Plant Sciences
Lines 11-12: “… A total of 5159 lysine 11 crotonylation sites in 2272 protein groups were identified. Twenty-seven motifs were found to be 12 the preferred amino acid sequences for crotonylation sites … ”
Query: Were there any motifs that occurred with higher frequency. If yes, then does it influence the activity?
Lines 21-22: “… HSP70 is a crucial protein involved in protecting plant cells and tissues from thermal or abiotic stress responses, ….”
Query: Does this lysine crotonylation gets influenced by strigolactone mediated regulatory systems?
Please Refer: Microorganisms 9:774. https://doi.org/10.3390/microorganisms9040774
Query: Are there potential implications of crotonylation of other amino-acids on the functioning of lysine crotonylations?
Query: Were lysine crotonylations limited only to leaves?
Query: Is this phenomenon a continuous process or age/phase-dependent?
Query: Did you observe a relation between the size of the motif and its functioning?
Query: Is this phenomenon related to respiration as well? Is there any potential benefit in manipulating it?
Minor issue: References have been poorly formatted.
Round 2
Reviewer 1 Report
Authors have improved considerably the manuscript . Please, revise the text carefully for text editing. Congratulations
